# A Single-Step, Sharpness-Aware Minimization is All You Need to Achieve Efficient and Accurate Sparse Training

**Jie Ji, Gen Li, Jingjing Fu, Fatemeh Afghah, Linke Guo, Xiaoyong Yuan, Xiaolong Ma**

Clemson University

jji@g.clemson.edu

## Abstract

Sparse training stands as a landmark approach in addressing the considerable training resource demands imposed by the continuously expanding size of Deep Neural Networks (DNNs). However, the training of a sparse DNN encounters great challenges in achieving optimal generalization ability despite the efforts from the state-of-the-art sparse training methodologies. To unravel the mysterious reason behind the difficulty of sparse training, we connect network sparsity with the structure of neural loss functions and identify that the cause of such difficulty lies in a chaotic loss surface. In light of such revelation, we propose $S^2$-SAM, characterized by a Single-step Sharpness-Aware Minimization that is tailored for Sparse training. For the first time, $S^2$-SAM innovates the traditional SAM-style optimization by approximating sharpness perturbation through prior gradient information, incurring *zero extra cost*. Therefore, $S^2$-SAM not only exhibits the capacity to improve generalization but also aligns with the efficiency goal of sparse training. Additionally, we study the generalization result of $S^2$-SAM and provide theoretical proof for convergence. Through extensive experiments, $S^2$-SAM demonstrates its universally applicable plug-and-play functionality, enhancing accuracy across various sparse training methods. Code available at https://github.com/jjsrf/SSAM-NEURIPS2024.

## 1 Introduction

The arrival of the Artificial General Intelligence (AGI) [1] era has urged an ever-expanding realm of artificial intelligence, bringing significant growth in deep neural networks (DNNs) depth and intricacy. The efficient training of DNN has thus emerged as an uttermost imperative, demanding immediate and concerted efforts.

To train an overparameterized, large-scale DNN efficiently while preserving its high accuracy, state-of-the-art literature has introduced sparse training [2–6] as a straightforward solution that reduces both parameter footprint and computation cost. With the presence of a large portion of zeros in the network, both forward and backward computation for training can be saved by skipping those zeros, as well as reducing the memory consumption since the zeros are not necessary to be stored. However, training a sparse neural network is difficult since the optimization under sparse regime readily converges to stationary points with a sub-optimal generalization accuracy (i.e., saddle points) [7]. Due to the difficulty of *directly* (i.e., without the time-consuming linear interpolation) assessing if the sparse solution is in the saddle point, finding workaround approaches for training a better sparse neural network is critical, especially for the practical usage in efficient learning [8–17].

To address the above difficulty, many efforts have been made. For instance, the Static Sparse Training (SST) such as the Lottery Ticket Hypothesis (LTH) [20], SNIP [21], GraSP [18] and SynFlow [22]

38th Conference on Neural Information Processing Systems (NeurIPS 2024).

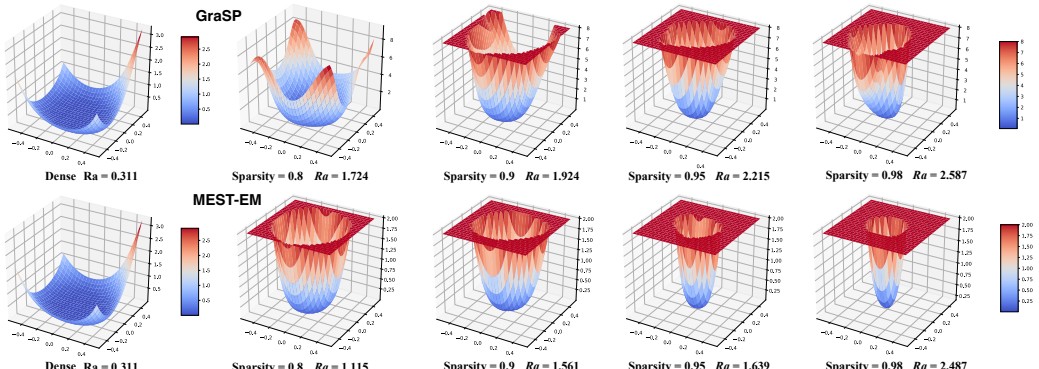

Figure 1: The loss surface visualization for training a sparse neural network using ResNet-32 on CIFAR-10. We select two representative sparse training methods [18, 3] and incorporate different levels of sparsity. We also quantify the loss surface behavior using coefficient $Ra$ [19] to evaluate sharpness. With increased sparsity, $Ra$ becomes larger, indicating sharper and steeper surface.

determine a static sparse pattern at training initialization. On the other hand, Dynamic Sparse Training (DST) such as SET [2], RigL [23], and MEST [3] iteratively updates the sparse topology during training to find a better sparse model. However, those methods either suffer from suboptimal generalization ability due to heuristic nature of their methodology or metric settings, or experience difficulties in parameter setting and dynamic sparsity scheduling. With the increase of the sparsity, these phenomenon become more severe.

We identify that the cause of the sparse network learning difficulties lies in the variable learning dynamics, which is closely related to the network topology. When incorporating sparsity into a neural network, the effective structure of the original network becomes narrower. According to the study of neural loss function structure [24], a wide network resulted in flat minima and wide regions of apparent convexity, which helps prevent the chaotic behavior that occurs during training. Evidently, higher sparsity indicates a narrower structure, which suggests more chaotic behavior is expected during training, thus degrading the accuracy. We perform different types of sparse training with different levels of sparsity and plot the loss surface in Figure 1 to demonstrate the convexity of the training behavior. According to the sharpness of the loss basin, we observe a transition from a smooth and close-to-convex surface to a steep one as sparsity is introduced and increased. Since higher sparsity levels lead to more chaotic behavior, in landscapes that are narrow and sharp with a large $Ra$ value, we might expect to encounter more chaotic training behavior that degrades generalization ability.

It seems that achieving coexistence of sparsity and good generalization ability during training is challenging. Therefore, we raise the following question: *is there a simple, effective method that can improve generalization ability of training a sparse neural network, without sacrificing efficiency (sparsity) and incurring zero extra cost?* We believe the answer to the above question lies in the sharpness of the loss surface. Inspired by the Sharpness-Aware Minimization (SAM) [25] technique that finds flatter minima that have uniformly low loss in nearby regions, we argue that such technique is especially suitable for sparse training since the steep loss surface induced by sparsity can be directly mitigated. However, to leverage sharpness for better generalization, SAM uses an additional full training step (i.e., forward and backward propagation) to quantify and evaluate loss on the summation of current weights and a constrained perturbation, which roughly doubles the computation of training. Such extra costs contradict the goal of sparse training. Although some literature [26, 27] have been proposed to reduce the computation cost for SAM, the computation is still significant, posing a roadblock to the wide application of SAM to the realm of sparse training.

In this paper, we propose a novel approach to achieve sharpness-aware minimization with *zero extra cost*, tailored for the sparse training regime to maintain efficiency while improving generalization ability. It is the *first time* that a **S**ingle-step **S**harpness-**A**ware **M**inimization is proposed for **S**parse training (S²-SAM). Different from the traditional two-step computation regime of SAM, S²-SAM uses a unique single-step approach that leverages sharpness and trains weights with one training step. Specifically, S²-SAM uses the weight gradients from the prior step to approximate the perturbation to the weights, thus solving the sharpness evaluation without performing an extra full training step. Therefore, S²-SAM incurs zero extra cost to achieve sharpness-aware training, which aligns with the

efficiency goal of sparse training. We also study the generalization result of S²-SAM and provide theoretical proof for convergence. We demonstrate that S²-SAM provides a straightforward plug-and-play functionality on variety of sparse training methods, significantly boosting their accuracy. Our contributions are summarized as follows:

- We identify that the difficulty of training a sparse neural network lies in the increasingly chaotic and steep loss surface when sparsity is introduced and increased.

- We develop a novel Single-step Sharpness-Aware Minimization technique tailored for Sparse training (S²-SAM), and it is the *first time* that a SAM-style optimization with *zero* extra computation cost has been proposed.

- We study the generalization result of S²-SAM and provide theoretical proof to demonstrate that the S²-SAM is guaranteed for convergence.

- Through systematic evaluations, we show that S²-SAM provides a plug-and-play functionality applied to a variety of sparse training methods, and consistently improves accuracy on different networks and datasets.

## 2 Proposed Method

### 2.1 Preliminary of Sharpness-Aware Minimization

SAM is an optimization technique designed to enhance neural network generalization and mitigate overfitting. It minimizes the maximum loss in a neighborhood around the current parameters, as opposed to solely focusing on the loss at the current point. This approach identifies flatter minima that have uniformly low loss in nearby regions, ultimately contributing to improved generalization performance.

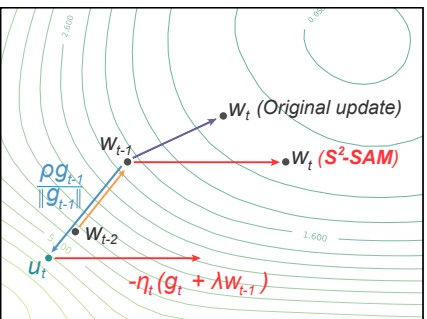

Figure 2: Illustration of the optimization mechanism of S²-SAM. The perturbation on the current weights is approximated by the weight gradients from prior step. Please see Section 2.2 for detailed discussion.

Specifically, consider a family of models parameterized by $\mathbf{w} \in \mathcal{W} \subseteq \mathbb{R}^d$; $L$ is the loss function, and $\mathcal{S}$ denotes the training dataset. SAM aims to minimize the following upper bound of the PAC-Bayesian generalization error: for any $\rho > 0$,

$$L(\mathbf{w}) \leq \max_{\|\epsilon\|_p \leq \rho} L_{\mathcal{S}}(\mathbf{w} + \epsilon) + \frac{\lambda}{2}\|\mathbf{w}\|^2. \tag{1}$$

To solve the above minimax problem, at each iteration $t$, SAM updates the following steps:

$$\begin{aligned} \epsilon_t &= \frac{\rho \cdot \text{sign}(\nabla L_{\mathcal{S}}(\mathbf{w}_{t-1})) |\nabla L_{\mathcal{S}}(\mathbf{w}_{t-1})|^{q-1}}{\left(\|\nabla L_{\mathcal{S}}(\mathbf{w}_{t-1})\|_q^q\right)^{1/p}}, \\ \mathbf{w}_t &= \mathbf{w}_{t-1} - \eta_t \left(\nabla L_{\mathcal{S}}\left(\mathbf{w}_{t-1} + \epsilon_t\right) + \lambda \mathbf{w}_{t-1}\right), \end{aligned} \tag{2}$$

where $1/p + 1/q = 1, \rho > 0$ is a hyperparameter, $\lambda > 0$ is the parameter for weight decay, and $\eta_t > 0$ is the learning rate. By setting $p = q = 2$ and introducing an intermediate variable $\mathbf{u}_t$, we have:

$$\mathbf{u}_t = \mathbf{w}_{t-1} + \frac{\rho \nabla L_{\mathcal{S}}\left(\mathbf{w}_{t-1}\right)}{\|\nabla L_{\mathcal{S}}\left(\mathbf{w}_{t-1}\right)\|}, \tag{3}$$

$$\mathbf{w}_t = \mathbf{w}_{t-1} - \eta_t \left(\nabla L_{\mathcal{S}}\left(\mathbf{u}_t\right) + \lambda \mathbf{w}_{t-1}\right). \tag{4}$$

### 2.2 The Proposed S²-SAM Method

As shown in Equation (2), SAM needs to compute the gradient twice at each iteration, involving additional computation costs. Further, the two-step gradient computation is not parallelizable, which presents challenges for deployment in large-scale training scenarios. Thus, we propose a new algorithm that only needs to compute the gradient *once* in each iteration. Different from prior efficient SAM works [26–28] aiming to reduce the computation by introducing periodically SAM steps or data

selection, our proposed framework S$^2$-SAM achieves *zero* extra computation cost while maintaining the improved generalization ability of sparse network training.

In Figure 2, we demonstrate that the perturbation on the current weights is approximated by the weight gradients $g_{t-1}$ from the prior step. The rational behind such design is that the gradient direction (i.e., $\mathbf{w}_{t-2}$ to $\mathbf{w}_{t-1}$) from prior step is optimizing the prior loss to a considerable extent, thus it can be used to represent a sharp direction among all perturbation directions at $t-1$. Since the loss surface of a sparse network is too chaotic, a perturbation with relatively high degree of sharpness (i.e., the prior gradient) can still find a neighborhood with near-maximum loss. The following equations specifying $g_{t-1}$ information are used to substitute the $\nabla L_\mathcal{S}(\mathbf{w}_{t-1})$ step in Equation (2). Thus, we only need to compute the gradient once in each iteration:

$$\mathbf{u}_t = \mathbf{w}_{t-1} + \frac{\rho \mathbf{g}_{t-1}}{\|\mathbf{g}_{t-1}\|}, \tag{5}$$

$$\mathbf{g}_t = \nabla L_\mathcal{S}(\mathbf{u}_t), \tag{6}$$

$$\mathbf{w}_t = \mathbf{w}_{t-1} - \eta_t(\mathbf{g}_t + \lambda \mathbf{w}_{t-1}). \tag{7}$$

*Remark* 1. The parameter $\rho$ can vary in terms of iteration $t : \rho_t = \sqrt{c/t}$, where $c > 0$ is a constant.

## 2.3 Generalization Analysis

In this section, we study the generalization result of S$^2$-SAM. First, we give some notations, where most of them are followed by [29, 30]. Let $\mathbf{w}_\mathcal{S} = \mathcal{A}(\mathcal{S})$ be a solution that generated by a random algorithm $\mathcal{A}$ based on dataset $\mathcal{S}$. Recall that problem (8)

$$\min_{\mathbf{w} \in \mathcal{W}} F_\mathcal{S}(\mathbf{w}) = \frac{1}{n} \sum_{i=1}^{n} \ell(\mathbf{y}_i, f(\mathbf{w}; \mathbf{x}_i)) \tag{8}$$

is called empirical risk minimization in the literature, and the true risk minimization is given by

$$\min_{\mathbf{w} \in \mathcal{W}} F(\mathbf{w}) := \mathrm{E}_{(\mathbf{x}, \mathbf{y})}[\ell(\mathbf{y}, f(\mathbf{w}; \mathbf{x}))]. \tag{9}$$

We define its optimal solution: $\mathbf{w}_* \in \arg\min_{\mathbf{w} \in \mathcal{W}} F(\mathbf{w})$. Then the excess risk bound (ERB) is defined as

$$\mathrm{E}_{\mathcal{A}, \mathcal{S}}[F(\mathbf{w}_\mathcal{S})] - F(\mathbf{w}_*). \tag{10}$$

It has been shown that the ERB can be upper bounded by optimization error and generalization error [29, 30]. We notice that there are several works [29, 30] studying the generalization result of SGD for non-convex setting under different conditions, such as bounded stochastic gradient $\|\nabla_\mathbf{w} \ell(\mathbf{y}, f(\mathbf{w}; \mathbf{x}))\| \leq G$ and decaying learning rate $\eta_t \leq \frac{c}{t}$ with a constant $c > 0$, where $t$ is the optimization iteration. In this paper, we are not interested in establishing a fast rate in ERB under different conditions, but we want to explore the generalization ability of S$^2$-SAM with the fewest possible modifications when building a bridge between theory and practice. For example, weight decay is a widely used trick when training deep neural networks. With the use of weight decay, the empirical risk minimization in practice becomes $\min_{\mathbf{w} \in \mathcal{W}} \left\{ \widehat{F}_\mathcal{S}(\mathbf{w}) := F_\mathcal{S}(\mathbf{w}) + \frac{\lambda}{2} \|\mathbf{w}\|^2 \right\}$. Then, we define some notations as follows. Specifically, let

$$\widehat{F}_\mathcal{S}(\mathbf{w}) = F_\mathcal{S}(\mathbf{w}) + \frac{\lambda}{2} \|\mathbf{w}\|^2 = \frac{1}{n} \sum_{i=1}^{n} \underbrace{\ell(\mathbf{y}_i, f(\mathbf{w}; \mathbf{x}_i)) + \frac{\lambda}{2} \|\mathbf{w}\|^2}_{\widehat{\ell}(\mathbf{y}_i, f(\mathbf{w}; \mathbf{x}_i))},$$

$$\widehat{F}(\mathbf{w}) = F(\mathbf{w}) + \frac{\lambda}{2} \|\mathbf{w}\|^2 \tag{11}$$

Followed by [30], we use the following decomposition of testing error:

$$\mathrm{E}_{\mathcal{A}, \mathcal{S}}[F(\mathbf{w}_\mathcal{S})] - \mathrm{E}_\mathcal{S}[F_\mathcal{S}(\mathbf{w}_\mathcal{S}^*)] \leq \mathrm{E}_\mathcal{S}[\mathrm{E}_\mathcal{A}[F_\mathcal{S}(\mathbf{w}_\mathcal{S}) - F_\mathcal{S}(\mathbf{w}_\mathcal{S}^*)]] + \mathrm{E}_{\mathcal{A}, \mathcal{S}}[F(\mathbf{w}_\mathcal{S}) - F_\mathcal{S}(\mathbf{w}_\mathcal{S})] \tag{12}$$

where the upper bound is the optimization error plus the generalization error.

Next, we present some notations and assumptions that will be used in the convergence analysis. Throughout this paper, we also make the following assumptions for solving the problem (8).

**Assumption 1.** Assume the following conditions hold: (i) The stochastic gradient of $F_{\mathcal{S}}(\mathbf{w})$ is unbiased, i.e., $\mathrm{E}_{(\mathbf{x},\mathbf{y})}[\nabla\ell(\mathbf{y},f(\mathbf{w};\mathbf{x}))] = \nabla F_{\mathcal{S}}(\mathbf{w})$, and the variance of stochastic gradient is bounded, i.e., there exists a constant $\sigma^2 > 0$, such that

$$\mathrm{E}_{(\mathbf{x},\mathbf{y})}\left[\|\nabla\ell(\mathbf{y},f(\mathbf{w};\mathbf{x})) - \nabla F_{\mathcal{S}}(\mathbf{w})\|^2\right] = \sigma^2.$$

(ii) $F_{\mathcal{S}}(\mathbf{w})$ is smooth with an L-Lipchitz continuous gradient, i.e., it is differentiable and there exists a constant $L > 0$ such that $\|\nabla F_{\mathcal{S}}(\mathbf{w}) - \nabla F_{\mathcal{S}}(\mathbf{u})\| \leq L\|\mathbf{w} - \mathbf{u}\|, \forall \mathbf{w}, \mathbf{u} \in \mathcal{W}$.

**Assumption 2.** There exists a constant $\mu > 0$ such that $2\mu(F_{\mathcal{S}}(\mathbf{w}) - F_{\mathcal{S}}(\mathbf{w}_{\mathcal{S}}^*)) \leq \|\nabla F_{\mathcal{S}}(\mathbf{w})\|^2, \forall \mathbf{w} \in \mathcal{W}$, where $\mathbf{w}_{\mathcal{S}}^* \in \min_{\mathbf{w}\in\mathcal{W}} F_{\mathcal{S}}(\mathbf{w})$ is a optimal solution (PL condition [31]).

Now consider that $\mathcal{A} = \mathrm{S}^2\text{-SAM}$. We define the gradient update rule $\mathcal{G}_{\widehat{\ell},\eta}$ as follows

$$\mathbf{u} = \mathbf{w} + \frac{\rho\nabla_{\mathbf{w}}\ell(\mathbf{y},f(\mathbf{w},\mathbf{x}))}{\|\nabla_{\mathbf{w}}\ell(\mathbf{y},f(\mathbf{w},\mathbf{x}))\|}, \tag{13}$$

$$\mathcal{G}_{\widehat{\ell},\eta}(\mathbf{w}) = \mathbf{w} - \eta\underbrace{(\nabla_{\mathbf{u}}\ell(\mathbf{y},f(\mathbf{u},\mathbf{x})) + \lambda\mathbf{w})}_{\nabla_{\mathbf{u}}\widetilde{\ell}(\mathbf{y},f(\mathbf{u},\mathbf{x}))}, \tag{14}$$

Then we have the following lemma, which is similar to Lemma 2.5 and Lemma 4.2 in [29] that use recursive definition through variable $\mathcal{G}'$ and $\mathbf{w}_t'$.

**Lemma 1.** *Assume that $\ell(\mathbf{y}, f(\mathbf{w}, \mathbf{x}))$ is L-smooth and B-Lipschitz. Let $\mathbf{w}_{t+1} = \mathcal{G}(\mathbf{w}_t)$ and another sequence $\mathbf{w}_{t+1}' = \mathcal{G}'(\mathbf{w}_t')$, then*

$$\|\mathbf{w}_{t+1} - \mathbf{w}_{t+1}'\| = \begin{cases} (1 + \eta L - \eta\lambda)\|\mathbf{w}_t - \mathbf{w}_t'\| + 2\eta\rho, & \mathcal{G} = \mathcal{G}' \\ (1 - \eta\lambda)\|\mathbf{w}_t - \mathbf{w}_t'\| + 2\eta B, & \mathcal{G} \neq \mathcal{G}' \end{cases}$$

Please see the proof of Lemma 1 in Appendix B.

**Theorem 1.** *Under Assumption 1, assume that $\ell(\mathbf{y}, f(\mathbf{w}, \mathbf{x}))$ is L-smooth and B-Lipschitz, suppose $\widehat{F}_{\mathcal{S}}(\mathbf{w})$ satisfies Assumption 2 and $\min_{\mathbf{w}\in\mathcal{W}} \widehat{F}_{\mathcal{S}}(\mathbf{w}) \leq F_{\mathcal{S}}(\mathbf{w}_{\mathcal{S}}^*) + \frac{\lambda}{2}\|\mathbf{w}_t\|^2$ with $\lambda = 2L$, where $\mathbf{w}_t$ is the intermediate solution of $\mathcal{A}$, then*

$$\mathrm{E}_{R,\mathcal{A},\mathcal{S}}[F(\mathbf{w}_R)] - \mathrm{E}_{\mathcal{S}}[F(\mathbf{w}_*)] \leq \frac{\widehat{F}_{\mathcal{S}}(\mathbf{w}_0)}{\mu\eta T} + \frac{\eta(L+\lambda)\sigma^2}{2\mu} + \frac{6B+1}{n}$$

*where $\mathcal{A}$ is SGD.*

The $\mathbf{w}_{\mathcal{S}}^* \in \min_{\mathbf{w}\in\mathcal{W}} F_{\mathcal{S}}(\mathbf{w})$ is an optimal solution, and we show that the generalization error is bounded. Based on Lemma 1, the proof of Theorem 1 is derived in Appendix C.

## 3 Experimental Results

In this section, we carry out comprehensive experiments to demonstrate how $\mathrm{S}^2$-SAM improves sparse training performance. We test $\mathrm{S}^2$-SAM on CIFAR-10/100 [34] with ResNet-32 [32] and VGG-19 [35], and we also perform experiments on ImageNet-1K [36] and ImageNet-C [37] based on ResNet-50 [38].

Following the recent developments in sparse training techniques, we apply $\mathrm{S}^2$-SAM on static sparse training such as LTH [20], SNIP [21], GraSP [18], as well as dynamic sparse training methods such as SET [2], DSR [33], RigL [23], MEST [3], CHEX [10] and Chase [39]. We apply $\mathrm{S}^2$-SAM to the official codes or published implementations to show performance gains. All of our experiments are performed on NVIDIA 4× A6000 GPUs. We repeat training experiments for 3 times and report the mean and standard deviation of the accuracy. For training throughput evaluation, we adopt the original settings of each baseline method and record the throughput on 4× A6000 GPUs.

Table 1: Test accuracy (%) of pruned ResNet-32 on CIFAR-10/100.

| Datasets | CIFAR-10 | | | CIFAR-100 | | |
|---|---|---|---|---|---|---|
| Pruning ratio | 90% | 95% | 98% | 90% | 95% | 98% |
| **ResNet-32** | 94.58 (Dense) | | | 74.89 (Dense) | | |
| LT [20] | 92.31 | 91.06 | 88.78 | 68.99 | 65.02 | 57.37 |
| LT+ S²-SAM (ours) | **92.58**±**0.07** (0.27↑) | **91.47**±**0.10** (0.41↑) | **89.35**±**0.11** (0.57↑) | **69.34**±**0.09** (0.35↑) | **65.45**±**0.11** (0.43↑) | **57.76**±**0.13** (0.39↑) |
| SNIP [21] | 92.59±0.10 | 91.01±0.21 | 87.51±0.31 | 68.89±0.45 | 65.02±0.69 | 57.37±1.43 |
| SNIP+ S²-SAM (ours) | **93.17**±**0.16** (0.58↑) | **91.59**±**0.22** (0.58↑) | **88.08**±**0.29** (0.57↑) | **69.33**±**0.28** (0.44↑) | **65.66**±**0.49** (0.64↑) | **58.25**±**0.77** (0.88↑) |
| GraSP [32] | 92.38±0.21 | 91.39±0.25 | 88.81±0.14 | 69.24±0.24 | 66.50±0.11 | 58.43±0.43 |
| GraSP+ S²-SAM (ours) | **92.87**±**0.14** (0.49↑) | **91.98**±**0.22** (0.59↑) | **89.66**±**0.29** (0.85↑) | **69.98**±**0.22** (0.74↑) | **67.12**±**0.18** (0.62↑) | **59.45**±**0.19** (1.02↑) |
| SET [2] | 92.30 | 90.76 | 88.29 | 69.66 | 67.41 | 62.25 |
| SET+ S²-SAM (ours) | **92.92**±**0.23** (0.62↑) | **91.50**±**0.19** (0.74↑) | **88.78**±**0.20** (0.49↑) | **70.23**±**0.20** (0.57↑) | **68.28**±**0.15** (0.87↑) | **63.56**±**0.19** (1.31↑) |
| DSR [33] | 92.97 | 91.61 | 88.46 | 69.63 | 68.20 | 61.24 |
| DSR+ S²-SAM (ours) | **93.49**±**0.21** (0.52↑) | **92.08**±**0.22** (0.47↑) | **89.11**±**0.17** (0.65↑) | **70.11**±**0.16** (0.48↑) | **68.87**±**0.16** (0.67↑) | **62.00**±**0.17** (0.76↑) |
| RigL [23] | 93.07 | 91.83 | 89.00 | 70.34 | 68.22 | 64.07 |
| RigL+ S²-SAM (ours) | **93.55**±**0.14** (0.48↑) | **92.11**±**0.21** (0.28↑) | **90.40**±**0.17** (1.40↑) | **72.38**±**0.11** (2.04↑) | **70.29**±**0.14** (2.07↑) | **64.98**±**0.06** (0.91↑) |
| RigL (ERK) [23] | 93.55 | 92.39 | 90.22 | 70.62 | 68.47 | 64.14 |
| RigL (ERK)+ S²-SAM (ours) | **93.75**±**0.19** (0.20↑) | **92.81**±**0.08** (0.42↑) | **91.16**±**0.11** (0.94↑) | **72.56**±**0.07** (1.94↑) | **70.33**±**0.10** (1.86↑) | **65.15**±**0.12** (1.01↑) |
| MEST (EM) [3] | 92.56±0.07 | 91.15±0.29 | 89.22±0.11 | 70.44±0.26 | 68.43±0.32 | 64.59±0.27 |
| MEST (EM) + S²-SAM (ours) | **93.43**±**0.12** (0.87↑) | **91.58**±**0.07** (0.43↑) | **91.22**±**0.14** (2.00↑) | **71.95**±**0.13** (1.51x↑) | **70.04**±**0.10** (1.61↑) | **65.69**±**0.34** (1.10↑) |
| MEST (EM&S) [3] | 93.27±0.14 | 92.44±0.13 | 90.51±0.11 | 71.30±0.31 | 70.36±0.05 | 67.16±0.25 |
| MEST (EM&S) + S²-SAM (ours) | **93.39**±**0.17** (0.12↑) | **92.97**±**0.17** (0.53↑) | **91.32**±**0.18** (0.81↑) | **72.74**±**0.08** (1.44↑) | **71.85**±**0.09** (1.49↑) | **69.13**±**0.20** (1.97↑) |

## 3.1 Accuracy Enhancement for State-of-the-art Sparse Training Methods

**CIFAR-10 and CIFAR-100.** We show S²-SAM is universally applicable on various sparse training methods. The results on ResNet-32 is shown in Table 1. Due to limited space, we demonstrate results of VGG-19 on CIFAR-10 in Table A.1 in the appendix. For each baseline method, we perform training with S²-SAM at 90%, 95% and 98% sparsity, and compare the accuracy with the original accuracy. We set hyper-parameter $\rho = 0.05$ for all experiments. From the comparison results, we can notice that all sparse training methods trained with S²-SAM obtain consistent and significant improvements, not only for static sparse training but also for dynamic sparse training. We stress that for all experiments, we only use one setting for S²-SAM, which indicates an easy implementation that has the potential for wide applicability. We also find that S²-SAM works better on high sparsity. For all sparse training with a 98% sparsity ratio, S²-SAM improves the original baseline accuracy by an average of 0.77% on CIFAR-10 and 1.12% on CIFAR-100, which shows S²-SAM is especially suitable for challenging tasks.

**ImageNet-1K.** We evaluate S²-SAM on ImageNet-1K dataset. The results are shown in Table 2. We report the top-1 accuracy of the sparse training results, as well the overall training cost with FLOPs. We set hyper-parameter $\rho = 1.0$ for all experiments. Compared to the baseline methods, our result shows higher test accuracy at the same computational cost. Similarly, S²-SAM also shows better gain on a higher sparsity ratio for most of the sparse training baselines, which indicates the consistency of using our methods on different scales of datasets. To make a fair comparison with different training recipes, we slightly scale up our training epochs to have the same or less overall training FLOPs to compare with longer training baselines, and we find out that S²-SAM shows stable improvement on those methods.

## 3.2 Sparse Training with Structured Sparsity

In this section, we evaluate how S²-SAM performs on structured sparse training methods. Note that all previous baselines use unstructured sparsity where the zeros are scattered in the network. Such unstructured sparse training shows the potential of training a sparse model, but cannot fully represent training acceleration ability. Therefore, we apply S²-SAM to two representative structured sparse trainings, CHEX [10] and Chase [39] which dynamically train sparse networks with *channel-level* sparsity. The results are shown in Table 3. From the results, we can see that the average accuracy improvement is around 0.5%. Considering structured (channel-wise) sparsity is one challenging sparsity type to achieve good accuracy, these results indicate S²-SAM works very well on structured sparse training methods.

## 3.3 Improvement on the Roughness of the Loss Surface

In this part, we plot the loss surface of the sparse training with and without S²-SAM. The comparisons are demonstrated in Figure 3, where we perform our experiments with three representative sparse training methods SNIP [21], GraSP [18] and MEST [3] at 90%, 95% and 98% sparsity ratios on

Table 2: Results of ResNet-50 on ImageNet-1K.

| Method | Sparsity Distribution | Top-1 Accuracy (%) | Training FLOPs | Inference FLOPs | Top-1 Accuary (%) | Training FLOPs | Inference FLOPs |
|---|---|---|---|---|---|---|---|
| ResNet-50 | dense | 76.9 | (×e18) | (×e9) | 76.9 | (×e18) | (×e9) |
| Sparsity | | 80% | | | 90% | | |
| LT [20] | | 72.6 | n/a | 2.7 | 70.1 | n/a | 1.7 |
| LT + S²-SAM (ours) | | **73.11±0.08** (0.51↑) | n/a | 2.7 | **70.78±0.05** (0.68↑) | n/a | 1.7 |
| SNIP [21] | non-uniform | 69.7 | 1.67 | 2.8 | 62.0 | 0.91 | 1.9 |
| SNIP + S²-SAM (ours) | | **70.55±0.05** (0.85↑) | 1.67 | 2.8 | **62.62±0.07** (0.62↑) | 0.91 | 1.9 |
| GraSP [32] | | 72.1 | 1.67 | 2.8 | 68.1 | 0.91 | 1.9 |
| GraSP + S²-SAM (ours) | | **72.66±0.10** (0.56↑) | 1.67 | 2.8 | **68.78±0.12** (0.68↑) | 0.91 | 1.9 |
| SET [2] | non-uniform | 72.9 | 0.74 | 1.7 | 69.6 | 0.10 | 0.1 |
| SET + S²-SAM (ours) | | **73.66±0.11** (0.76↑) | 0.74 | 1.7 | **70.41±0.08** (0.81↑) | 0.10 | 0.1 |
| DSR [33] | | 73.3 | 1.28 | 3.3 | 71.6 | 0.96 | 2.5 |
| DSR + S²-SAM (ours) | | **74.08±0.17** (0.78↑) | 1.28 | 3.3 | **72.32±0.13** (0.72↑) | 0.96 | 2.5 |
| RigL [23] | | 74.6 | 0.74 | 1.7 | 72.0 | 0.39 | 0.9 |
| RigL + S²-SAM (ours) | | **75.39±0.12** (0.79↑) | 0.74 | 1.7 | **72.44±0.06** (0.44↑) | 0.39 | 0.9 |
| MEST (EM) [3] | uniform | 75.7 | 1.10 | 1.7 | 73.6 | 0.48 | 0.9 |
| MEST (EM) + S²-SAM (ours) | | **76.35±0.02** (0.65↑) | 1.10 | 1.7 | **74.58±0.03** (0.98↑) | 0.48 | 0.9 |
| MEST (EM&S) [3] | | 75.7 | 1.27 | 1.7 | 75.0 | 0.65 | 0.9 |
| MEST (EM&S) + S²-SAM (ours) | | **76.44±0.06** (0.74↑) | 1.27 | 1.7 | **75.36±0.04** (0.36↑) | 0.65 | 0.9 |
| Top-KAST [40] | | - | - | - | 73.0 | 0.63 | 0.9 |
| Top-KAST + S²-SAM (ours) | | - | - | - | **73.82±0.17** (0.82↑) | 0.63 | 0.9 |
| MEST₁.₇ₓ [3] | uniform | 76.7 | 1.84 | 1.7 | 75.9 | 0.80 | 0.9 |
| MEST₁.₇ₓ + S²-SAM (ours) | | **77.73±0.11** (1.03↑) | 1.84 | 1.7 | **76.82±0.12** (0.92↑) | 0.80 | 0.9 |
| RigL₅ₓ [23] | | 76.6 | 3.71 | 1.7 | 75.7 | 1.95 | 0.9 |
| RigL₅ₓ + S²-SAM (ours) | | **77.72±0.05** (1.12↑) | 3.71 | 1.7 | **76.88±0.13** (1.18↑) | 1.95 | 0.9 |

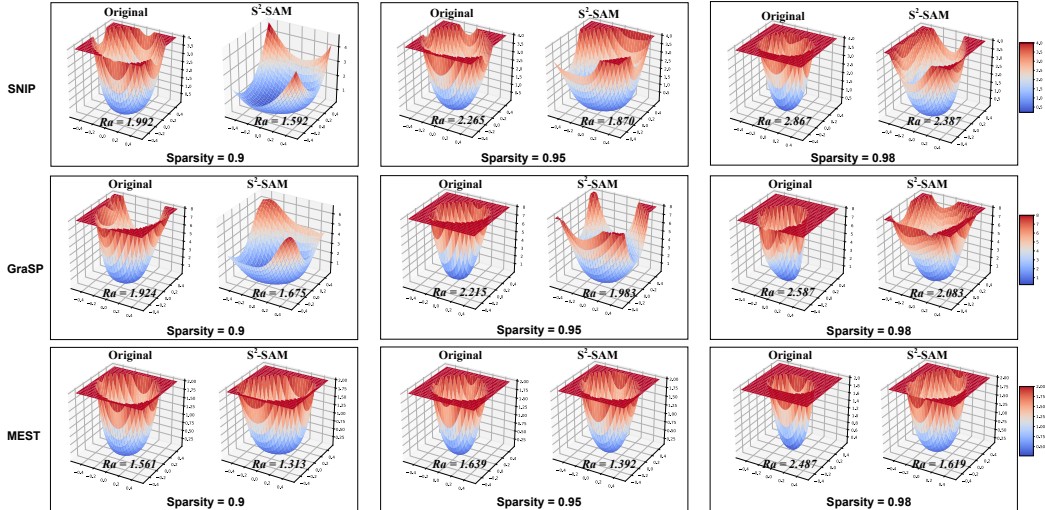

Figure 3: Loss surface sharpness comparison of different sparse training methods with original training and with $S^2$-SAM. We also quantitatively evaluate the coefficient $Ra$. Using $S^2$-SAM compared to the original method results in a smaller $Ra$, indicating a wider and smoother loss surface, which suggests improved generalization ability.

CIFAR-10 dataset with ResNet-32. We observe that as sparsity increases, suggesting more chaotic training behavior, the coefficient $Ra$ also increases, indicating a steeper loss surface and a narrower basin. While with $S^2$-SAM, we can see that under the same sparsity, the basin of the loss surface widens and enlarges as $Ra$ values decrease, suggesting a smoother loss trajectory during training of a sparse network.

## 3.4 Training Speed Comparison on GPU

$S^2$-SAM is a highly efficient approach to optimizing the sharpness of the loss surface, which incurs zero extra cost for achieving its functionality. Compared to the traditional SAM, $S^2$-SAM uses fewer computations and has better training performance on GPUs. In Table 4, we obtain the computation cost as well as the training speed in terms of throughput (i.e., imgs/sec) on $4\times$ NVIDIA A6000 GPUs

using ResNet-50 on the ImageNet-1K dataset. We record the throughput for different sparse training methods with their original training, with SAM [25], and with S$^2$-SAM. From the results, we can see that S$^2$-SAM achieves the same training throughput (with only negligible $< 20$ imgs/s decrease due to processing) as the baseline methods, where no sharpness optimization is involved. Compared to original SAM, we can see that training throughput with SAM is *less than half* of the ones that train with S$^2$-SAM, which aligns with the observation that SAM uses *twice the computation* cost of S$^2$-SAM or original training. Although SAM yields slightly better accuracy, the associated training cost outweighs the benefits, rendering it impractical.

Table 3: Accuracy of S$^2$-SAM on structured sparse training CHEX [10] and Chase [39].

| Methods | Networks | Training | FLOPs | Accuracy (%) |
|---|---|---|---|---|
| CHEX | ResNet-34 | Original | 2.0G | 73.50 |
| | | S$^2$-SAM | 2.0G | **73.94** (0.44↑) |
| | ResNet-50 | Original | 1.0G | 76.00 |
| | | S$^2$-SAM | 1.0G | **76.51** (0.51↑) |
| Chase | ResNet-34 | Original* | 2.1G | 72.34 |
| | | S$^2$-SAM | 2.1G | **72.77** (0.43↑) |
| | ResNet-50 | Original | 1.3G | 75.62 |
| | | S$^2$-SAM | 1.3G | **76.17** (0.55↑) |

* ResNet-34 original result of Chase is our own implementation.

Table 4: Training speed of SAM [25] and S$^2$-SAM for different sparse training at 90% sparsity.

| Methods | Training | Accuracy (%) | Throughput (↑) |
|---|---|---|---|
| GraSP | Original | 68.10 | **2148** imgs/s |
| | SAM | **68.95** | 1021 imgs/s |
| | S$^2$-SAM | 68.78 | 2132 imgs/s |
| RigL | Original | 72.00 | **3133** imgs/s |
| | SAM | **72.75** | 1508 imgs/s |
| | S$^2$-SAM | 72.44 | 3098 imgs/s |
| MEST (EM) | Original | 73.60 | **2981** imgs/s |
| | SAM | **74.88** | 1398 imgs/s |
| | S$^2$-SAM | 74.58 | 2977 imgs/s |

## 3.5 Robustness Improvement by S$^2$-SAM

Since sparse training approaches have potentially high usage in practical scenarios, the system's robustness against perturbation (e.g., inclement weather conditions for image/video tasks) is usually critical. We perform experiments to evaluate the sparse model (80% sparsity) robustness against perturbations in Table 5. Our intuition is that the model trained with its sharpness optimized has a wider loss basin, which indicates higher endurance on perturbations since the loss won't change much when it is located in a wide and flat region. We adopt ImageNet-C [37] which contains a test set with the same images as ImageNet-1K but with nineteen types of corruptions applied with five different levels of severity. We train the sparse networks using different methods on ImageNet-1K, and then test their accuracy on ImageNet-C. We report test accuracy for both datasets. We can see that when the sparse model is trained using the original method, the accuracy of ImageNet-C is significantly lower than the accuracy of the original ImageNet-1K test set (around 35% lower). With S$^2$-SAM, the test accuracy on ImageNet-C shows promising improvement. The robust accuracy improves by an average of 3.23%.

Table 5: Testing accuracy on ImageNet-C test set. We compare the results with and without S$^2$-SAM using 80% sparsity.

| Methods | ImageNet-1K Accuracy (%) | ImageNet-C Accuracy (%) |
|---|---|---|
| SNIP | 69.70 | 31.12 |
| SNIP + S$^2$-SAM | **70.55** (0.85↑) | **34.87** (3.75↑) |
| GraSP | 72.10 | 32.24 |
| GraSP + S$^2$-SAM | **72.66** (0.56↑) | **35.17** (2.93↑) |
| MEST (EM) | 75.70 | 33.87 |
| MEST (EM) + S$^2$-SAM | **76.35** (0.65↑) | **36.98** (3.11↑) |
| RigL | 74.60 | 33.68 |
| RigL + S$^2$-SAM | **75.39** (0.79↑) | **36.80** (3.12↑) |

Table 6: Testing accuracy on dense model training. We compare original training with S$^2$-SAM in same settings.

| Networks | Params. Count | Original Accuracy (%) | S$^2$-SAM Accuracy (%) |
|---|---|---|---|
| CIFAR-10 | | | |
| ResNet-32 | 1.86M | 94.58 | **94.99** (0.41↑) |
| MobileNet-V2 | 2.30M | 94.13 | **94.55** (0.42↑) |
| VGG-19 | 20.03M | 94.21 | **94.48** (0.27↑) |
| ImageNet-1K | | | |
| EfficientNet-B0 | 5.30M | 76.54 | **77.10** (0.56↑) |
| ResNet-34 | 21.80M | 74.09 | **74.58** (0.49↑) |
| ResNet-50 | 25.50M | 76.90 | **77.32** (0.42↑) |

## 3.6 Applying S$^2$-SAM to Dense Model Training

We also apply S$^2$-SAM to the dense model training. The results are shown in Table 6. We test on two datasets CIFAR-10 and ImageNet-1K with two additional networks MobileNet-V2 [41] and EfficientNet-B0 [42]. For ResNet-32 and VGG-19 on CIFAR-10, we follow the settings in [3] and train for 160 epochs, and for MobileNet-V2, we use the 1.0 width version and train for 350 epochs

for better convergence [43]. For ResNet family on ImageNet-1K, we follow the setting in [3] and train for 150 epochs, and we train EfficientNet-B0 for 350 epochs for better convergence [44]. From the results, we can see that $S^2$-SAM still improves the accuracy of the dense networks. We find out that the less the parameter count of the network, the better effectiveness of $S^2$-SAM it achieves. This phenomenon proves our finding that a narrower network is harder to train, and $S^2$-SAM is an effective solution for better generalization. We must also stress that $S^2$-SAM is focusing on sparse neural network training, and we will leave the study of $S^2$-SAM on dense model training for our future research.

## 4 Related Works

**Static Sparse Training** Static sparse training determines the structure of the sparse network through the application of a pruning algorithm in the early stages of training. The lottery ticket hypothesis (LTH) [20, 45, 46] uses iterative magnitude-based pruning (IMP) to find a subnetwork that can be trained from scratch without losing accuracy. SNIP [21], GraSP [18], SynFlow [22] determine a static sparse pattern at training initialization by obtaining gradient information with a few iterations of dense training. FISH [47] acquires fixed subnetwork by pre-computing a sparse mask using Fisher information.

**Dynamic Sparse Training** Dynamic sparse training starts with a randomly selected sparse network structure and adapts it throughout the training process in an effort to find a better sparse structure. Sparse Evolutionary Training (SET) [2] prunes small magnitude weights and grows back randomly at the end of each training epoch. Deep R [48] uses a combination of stochastic parameter updates and dynamic sparse parameterization for training. Dynamic Sparse Reparameterization (DSR) [33] proposes to redistribute parameters between layers during training. Sparse Networks from Scratch (SNFS) [49] creates the sparse momentum algorithm, which finds the layers and weights that effectively reduce the error by using exponentially smoothed gradients (momentum). In RigL [23], the gradients of all the weights are computed when the model needs to be updated to grow new connections. Top-KAST [40] proposes a scalable and performant sparse-to-spars DST framework for maximum efficacy. Powerpropagation [50] suggests a novel neural network weight parameterization that largely preserves low-magnitude parameters from learning. ITOP [51] investigates the underlying DST mechanism and finds that the advantages of DST result from a time-based search for all potential factors. MEST [3] designs a memory-economic sparse training framework targeting accurate and fast execution on edge devices. By co-training dense and sparse models, AD/AC [52] suggests a technique that, at the conclusion of training, produces precise sparse–dense model pairings. Chase [39] dynamically translates the unstructured sparsity into channel-level sparsity to achieve direct speedup on GPU.

**Sharpness-Aware Minimization** Sharpness-Aware Minimization was first introduced in [25]. This optimization technique is designed to identify flatter minima characterized by consistently low loss in neighboring regions, aiming to enhance the model generalization capability during training. Nevertheless, the computational cost of Sharpness-Aware Minimization is doubled due to its two-step gradient computation regime, presenting challenges for deployment in large-scale training scenarios. To reconcile such, ESAM [26] introduces two novel and efficient training strategies: stochastic weight perturbation and sharpness-sensitive data selection, enhancing the efficiency of the SAM process without compromising its generalization performance. LookSAM [27] only periodically calculates the inner gradient ascent to significantly reduce the additional training cost of SAM. SAF [28] introduces a novel trajectory loss based on KL-divergence to measure the rate of change in training loss along the model update trajectory and replace the SAM sharpness measure. CrAM [53] optimizes over the compression projection applied to the intermediate model at every training step, which results in a compressible models that can be pruned in an on-shot manner after training. However, all those methods need extra computation cost, and not target on sparse training method.

## 5 Conclusion

In this paper, we propose a novel Single-step Sharpness-Aware Minimization that is tailored for Sparse training ($S^2$-SAM), which revolutionizes the originally computation-intensive sharpness-aware optimization into a highly efficient tool with zero extra cost. In light of the improved optimization trajectory in the loss surface, $S^2$-SAM successfully enhances the accuracy of the sparse network

training, as well as the robustness of the sparse model in practical scenarios. $S^2$-SAM offers seamless plug-and-play functionality, showcasing its potential for widespread applicability in the evolving landscape of efficient training. The research is inherently scientific, and we anticipate no adverse societal impact stemming from its findings.

## Acknowledgments

This work is partly supported by the National Science Foundation CCF-2427875, CCF-2312616, CNS-2232048, CNS-2008049, CNS-2431440, National Aeronautics and Space Administration (NASA) 80NSSC23K1393, and the Army Research Office W911NF-24-1-0044. The views and conclusions contained in this document are those of the authors and should not be interpreted as representing the official policies, either expressed or implied, of NSF, NASA, the Army Research Office or the U.S. Government. The U.S. Government is authorized to reproduce and distribute reprints for Government purposes notwithstanding any copyright notation herein.

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

# Appendix

## A  Results of VGG-19 on CIFAR-10/100

We test $S^2$-SAM on VGG-19 using CIFAR-10 and CIFAR-100. The results are shown in Table A.1. We demonstrate results on both uniform and ERK distributions, and achieve SOTA results on CIFAR-10/100.

Table A.1: Test accuracy (%) of pruned VGG-19 on CIFAR-10/100.

| Datasets | CIFAR-10 | | | CIFAR-100 | | |
|---|---|---|---|---|---|---|
| Pruning ratio | 90% | 95% | 98% | 90% | 95% | 98% |
| **VGG-19** | 94.20 | | | 74.17 | | |
| LT [20] | 93.51 | 92.92 | 92.34 | 72.78 | 71.44 | 68.95 |
| LT+ $S^2$-SAM (ours) | **93.82±0.12** (0.31↑) | **93.41±0.11** (0.49↑) | **92.92±0.14** (0.58↑) | **73.08±0.08** (0.30↑) | **71.69±0.10** (0.25↑) | **69.78±0.17** (0.83↑) |
| SNIP [21] | 93.63±0.06 | 93.43±0.20 | 92.05±0.28 | 72.84±0.22 | 71.83±0.23 | 58.46±1.10 |
| SNIP+ $S^2$-SAM (ours) | **93.91±0.11** (0.28↑) | **93.91±0.21** (0.48↑) | **92.86±0.28** (0.81↑) | **73.33±0.21** (0.49↑) | **72.41±0.35** (0.58↑) | **60.12±0.42** (1.66↑) |
| GraSP [32] | 93.30±0.14 | 93.43±0.18 | 92.19±0.12 | 71.95±0.18 | 71.23±0.12 | 68.90±0.41 |
| GraSP+ $S^2$-SAM (ours) | **93.88±0.11** (0.58↑) | **93.75±0.21** (0.32↑) | **92.95±0.23** (0.76↑) | **72.44±0.27** (0.49↑) | **72.91±0.25** (1.68↑) | **69.98±0.31** (1.08↑) |
| SET [2] | 92.46 | 91.73 | 89.18 | 72.36 | 69.81 | 65.94 |
| SET+ $S^2$-SAM (ours) | **92.97±0.22** (0.51↑) | **92.55±0.20** (0.82↑) | **90.18±0.19** (1.00↑) | **72.65±0.14** (0.29↑) | **70.11±0.14** (0.30↑) | **66.76±0.17** (0.82↑) |
| DSR [33] | 93.75 | 93.86 | 93.13 | 72.31 | 71.98 | 70.70 |
| DSR+ $S^2$-SAM (ours) | **94.24±0.13** (0.49↑) | **94.27±0.15** (0.41↑) | **93.81±0.27** (0.68↑) | **72.78±0.20** (0.47↑) | **72.55±0.22** (0.57↑) | **71.69±0.24** (0.99↑) |
| RigL [23] | 93.12 | 92.43 | 90.65 | 71.14 | 69.02 | 64.87 |
| RigL+ $S^2$-SAM (ours) | **93.61±0.16** (0.49↑) | **92.87±0.21** (0.44↑) | **91.88±0.21** (1.23↑) | **71.98±0.18** (0.84↑) | **70.16±0.18** (1.14↑) | **66.01±0.25** (1.14↑) |
| RigL (ERK) [23] | 93.77 | 92.75 | 90.87 | 71.34 | 69.21 | 65.02 |
| RigL (ERK)+ $S^2$-SAM (ours) | **94.07±0.21** (0.30↑) | **93.33±0.12** (0.58↑) | **91.79±0.12** (0.92↑) | **72.12±0.20** (0.78↑) | **69.95±0.18** (0.74↑) | **66.13±0.19** (1.11↑) |
| MEST (EM) [3] | 93.07±0.36 | 92.59±0.41 | 90.55±0.44 | 71.23±0.37 | 69.08±0.41 | 64.92±0.34 |
| MEST (EM) + $S^2$-SAM (ours) | **93.76±0.13** (0.69↑) | **93.18±0.12** (0.59↑) | **91.86±0.23** (1.31↑) | **71.93±0.11** (0.70↑) | **69.98±0.10** (0.90↑) | **66.08±0.16** (1.16↑) |
| MEST (EM&S) [3] | 93.61±0.36 | 93.46±0.41 | 92.30±0.44 | 72.52±0.37 | 71.21±0.41 | 69.02±0.34 |
| MEST (EM&S) + $S^2$-SAM (ours) | **94.51±0.11** (0.90↑) | **93.98±0.11** (0.52↑) | **91.55±0.14** (0.75↑) | **72.90±0.17** (0.38↑) | **72.42±0.09** (1.21↑) | **71.01±0.13** (1.99↑) |

## B  Proof of Lemma 1

*Proof.* We consider two cases.

(1) When $\mathcal{G} = \mathcal{G}'$ and $\ell(\cdot, f(\mathbf{w}; \cdot))$ is $L$-smooth, then

$$\left\| \mathbf{w}_{t+1} - \mathbf{w}'_{t+1} \right\| \leq (1 - \eta\lambda) \left\| \mathbf{w}_t - \mathbf{w}'_t \right\| + \eta \left\| \nabla_{\mathbf{u}} \ell\left(\mathbf{y}_t, f\left(\mathbf{u}_t; \mathbf{x}_t\right)\right) - \nabla_{\mathbf{u}} \ell\left(\mathbf{y}_t, f\left(\mathbf{u}'_t; \mathbf{x}_t\right)\right) \right\|$$
$$\leq (1 - \eta\lambda) \left\| \mathbf{w}_t - \mathbf{w}'_t \right\| + \eta L \left\| \mathbf{u}_t - \mathbf{u}'_t \right\|$$
$$\leq (1 + \eta L - \eta\lambda) \left\| \mathbf{w}_t - \mathbf{w}'_t \right\| + 2\eta\rho.$$

(2) When $\mathcal{G} \neq \mathcal{G}'$ and $\ell(\cdot, f(\mathbf{w}; \cdot))$ is $B$-Lipschitz, then

$$\left\| \mathbf{w}_{t+1} - \mathbf{w}'_{t+1} \right\| \leq (1 - \eta\lambda) \left\| \mathbf{w}_t - \mathbf{w}'_t \right\| + \eta \left\| \nabla_{\mathbf{u}} \ell\left(\mathbf{y}_t, f\left(\mathbf{u}_t; \mathbf{x}_t\right)\right) - \nabla_{\mathbf{u}} \ell\left(\mathbf{u}'_t, f\left(\mathbf{w}'_t; \mathbf{x}'_t\right)\right) \right\|$$
$$\leq (1 - \eta\lambda) \left\| \mathbf{w}_t - \mathbf{w}'_t \right\| + 2\eta B.$$

$\square$

## C  Proof of Theorem 1

*Proof.* By Theorem 3.2 of [29], we have

$$\Delta_{t+1} := \mathrm{E}\left[\left\| \mathbf{w}_{t+1} - \mathbf{w}'_{t+1} \right\|\right]$$
$$\overset{(a)}{\leq} \left(1 - \frac{1}{n}\right)(1 + \eta L - \eta\lambda)\Delta_t + \left(1 - \frac{1}{n}\right) 2\eta\rho + \frac{1}{n}\left((1 - \eta\lambda)\Delta_t + 2\eta B\right)$$
$$= \left[1 + \left(1 - \frac{1}{n}\right)\eta L - \eta\lambda\right]\Delta_t + \frac{2\eta B}{n}$$
$$\overset{(b)}{\leq} (1 - \eta L)\Delta_t + \frac{2\eta B + 2(n-1)\eta\rho}{n}$$
$$\overset{(c)}{\leq} \frac{2\eta B + 2(n-1)\eta\rho}{n}\sum_{i=0}^{t_0}(1 - \eta L)^i \overset{(d)}{\leq} \frac{2B + 2(n-1)\rho}{nL}, \tag{15}$$

where (a) uses Lemma 1; (b) uses $\lambda = 2L$; (c) uses $\Delta_{t_0} = 0$; (d) uses $\eta L < 1$. Then, by Lemma 3.11 of [29] and $\widehat{\ell}$ is $(L + \lambda)$-smooth, we have

$$\mathrm{E}\left[\widehat{\ell}(\mathbf{w}_t; \mathbf{z}) - \widehat{\ell}(\mathbf{w}'_t; \mathbf{z})\right] \leq \frac{t_0}{n} + \frac{2(B + (n-1)\rho)(L+\lambda)}{nL} \leq \frac{6(B + (n-1)\rho) + 1}{n} \qquad (16)$$

where the last inequality holds by selecting $t_0 = 1$ and $\lambda = 2L$. Then, by Theorem 2.2 of [29]

$$\mathrm{E}_{\mathcal{A},\mathcal{S}}\left[\widehat{F}(\mathbf{w}_t) - \widehat{F}_{\mathcal{S}}(\mathbf{w}_t)\right] \leq \frac{6(B + (n-1)\rho) + 1}{n} \qquad (17)$$

which is the generalization error. Next, we will bound the optimization error. Since $\widehat{F}_{\mathcal{S}}$ is $(L + \lambda)$-smooth and satisfies $\mu$-PL condition, then follow the standard analysis, we have

Since $\widehat{F}_{\mathcal{S}}$ is $(L + \lambda)$-smooth, we have

$$\widehat{F}_{\mathcal{S}}(\mathbf{w}_t) - \widehat{F}_{\mathcal{S}}(\mathbf{w}_{t-1})$$

$$\leq \left\langle \nabla \widehat{F}_{\mathcal{S}}(\mathbf{w}_{t-1}), \mathbf{w}_t - \mathbf{w}_{t-1} \right\rangle + \frac{L+\lambda}{2} \|\mathbf{w}_t - \mathbf{w}_{t-1}\|^2$$

$$= -\eta \left\langle \nabla \widehat{F}_{\mathcal{S}}(\mathbf{w}_{t-1}), \nabla_{\mathbf{u}}\widetilde{\ell}(\mathbf{y}, f(\mathbf{u}_t, \mathbf{x})) \right\rangle + \frac{\eta^2(L+\lambda)}{2} \left\|\nabla_{\mathbf{u}}\widetilde{\ell}(\mathbf{y}, f(\mathbf{u}_t, \mathbf{x}))\right\|^2$$

$$\leq -\eta \left\langle \nabla \widehat{F}_{\mathcal{S}}(\mathbf{w}_{t-1}), \nabla_{\mathbf{w}}\widehat{\ell}(\mathbf{y}, f(\mathbf{w}_{t-1}, \mathbf{x})) \right\rangle - \eta \left\langle \nabla \widehat{F}_{\mathcal{S}}(\mathbf{w}_{t-1}), \nabla_{\mathbf{u}}\widetilde{\ell}(\mathbf{y}, f(\mathbf{u}_t, \mathbf{x})) - \nabla_{\mathbf{w}}\widehat{\ell}(\mathbf{y}, f(\mathbf{w}_{t-1}, \mathbf{x})) \right\rangle$$
$$\qquad\qquad (18)$$

$$+ \frac{\eta^2(L+\lambda)B^2}{2}$$

$$\leq -\eta \left\langle \nabla \widehat{F}_{\mathcal{S}}(\mathbf{w}_{t-1}), \nabla_{\mathbf{w}}\widehat{\ell}(\mathbf{y}, f(\mathbf{w}_{t-1}, \mathbf{x})) \right\rangle + \frac{\eta}{2}\left\|\nabla \widehat{F}_{\mathcal{S}}(\mathbf{w}_{t-1})\right\|^2 + \frac{\eta\rho^2}{2} + \frac{\eta^2(L+\lambda)B^2}{2}$$
$$\qquad\qquad (19)$$

where the last inequality is due to $\ell(\mathbf{y}, f(\mathbf{w}, \mathbf{x}))$ is $B$-Lipschitz. Therefore, we get

$$\frac{1}{T}\sum_{t=0}^{T-1} \mathrm{E}\left[\left\|\nabla \widehat{F}_{\mathcal{S}}(\mathbf{w}_t)\right\|^2\right] \leq \frac{2\widehat{F}_{\mathcal{S}}(\mathbf{w}_0)}{\eta} + \rho + \eta(L+\lambda)B^2. \qquad (20)$$

Since $\widehat{F}_{\mathcal{S}}$ satisfies $\mu$-PL condition, then

$$\frac{1}{T}\sum_{t=0}^{T-1} \mathrm{E}_{\mathcal{S}}\left[\mathrm{E}_{\mathcal{A}}\left[\widehat{F}_{\mathcal{S}}(\mathbf{w}_t) - \widehat{F}_{\mathcal{S}}(\widehat{\mathbf{w}}_{\mathcal{S}}^*)\right]\right] \leq \frac{1}{2\mu T}\sum_{t=0}^{T-1} \mathrm{E}\left[\left\|\nabla \widehat{F}_{\mathcal{S}}(\mathbf{w}_t)\right\|^2\right]$$

$$\leq \frac{\widehat{F}_{\mathcal{S}}(\mathbf{w}_0)}{\mu\eta T} + \frac{\rho + \eta(L+\lambda)B^2}{2\mu} \qquad (21)$$

Then by (12), (17) and (21), we get

$$\frac{1}{T}\sum_{t=0}^{T-1} \mathrm{E}_{\mathcal{A},\mathcal{S}}\left[\widehat{F}(\mathbf{w}_t)\right] - \mathrm{E}_{\mathcal{S}}\left[\widehat{F}_{\mathcal{S}}(\widehat{\mathbf{w}}_{\mathcal{S}}^*)\right] \leq \frac{\widehat{F}_{\mathcal{S}}(\mathbf{w}_0)}{\mu\eta T} + \frac{\rho + \eta(L+\lambda)B^2}{2\mu} + \frac{6(B + (n-1)\rho) + 1}{n}$$
$$\qquad\qquad (22)$$

By using the conditions that $\min_{\mathbf{w}\in\mathcal{W}}\widehat{F}_{\mathcal{S}}(\mathbf{w}) \leq F_{\mathcal{S}}(\mathbf{w}_{\mathcal{S}}^*) + \frac{\lambda}{2}\|\mathbf{w}_t\|^2$, definitions (11), we get

$$F(\mathbf{w}_t) - F_{\mathcal{S}}(\mathbf{w}_{\mathcal{S}}^*) \leq \widehat{F}(\mathbf{w}_t) - \widehat{F}_{\mathcal{S}}(\widehat{\mathbf{w}}_{\mathcal{S}}^*) \qquad (23)$$

Therefore, we have the following inequality by (22) and (23):

$$\frac{1}{T}\sum_{t=0}^{T-1} \mathrm{E}_{\mathcal{A},\mathcal{S}}[F(\mathbf{w}_t)] - \mathrm{E}_{\mathcal{S}}[F_{\mathcal{S}}(\mathbf{w}_{\mathcal{S}}^*)] \leq \frac{\widehat{F}_{\mathcal{S}}(\mathbf{w}_0)}{\mu\eta T} + \frac{\rho + \eta(L+\lambda)B^2}{2\mu} + \frac{6(B + (n-1)\rho) + 1}{n}$$
$$\qquad\qquad (24)$$

By Lemma 5.1 of [29], (24) implies the ERB is

$$\mathrm{E}_{R,\mathcal{A},\mathcal{S}}[F(\mathbf{w}_R)] - \mathrm{E}_{\mathcal{S}}[F(\mathbf{w}_*)] \leq \frac{\widehat{F}_{\mathcal{S}}(\mathbf{w}_0)}{\mu\eta T} + \frac{\rho + \eta(L+\lambda)B^2}{2\mu} + \frac{6(B + (n-1)\rho) + 1}{n} \qquad (25)$$

By setting $\eta = O(1/\sqrt{n})$ and $T = O(n)$ then $\mathrm{E}_{R,\mathcal{A},\mathcal{S}}[F(\mathbf{w}_R)] - F(\mathbf{w}_*) \leq O(1/\sqrt{n})$, where $\mathcal{A}$ is $S^2$-SAM. $\qquad\square$

