# OpenReview forum: "A Single-Step, Sharpness-Aware Minimization is All You Need to Achieve Efficient and Accurate Sparse Training"
_NeurIPS.cc/2024/Conference — NeurIPS 2024 poster_

### Official Review · Reviewer_UgUE · 2024-06-24

**Soundness:** 3
**Presentation:** 3
**Contribution:** 3
**Rating:** 7
**Confidence:** 4

**Summary:**

This paper presents a sparse training method, $S^2$-SAM, which applies sharpness-aware minimization to sparse training. The authors demonstrate that sparsity during training leads to a sharper (to use the authors' words, "more chaotic") loss surface, something that can be mitigated by a variant of sharpness-aware training. Additionally, the authors avoid the extra gradient computation of SAM by reusing the gradient from the previous step as a proxy. The authors demonstrate theoretically that the error from $S^2$-SAM is bounded, and empirically by testing their algorithm with a number of sparse training methods, as well as with dense training.

**Strengths:**

The method is practical and efficient, and achieves very good experimental results. I appreciated the clock-time comparison in addition to the usual FLOPs. Overall, the paper is quite convincing, and I personally plan to give the method a try in my own work.

**Weaknesses:**

The main experimental weakness of the paper is restricting the method to sparsities no higher than 90 on ImageNet and not doing a full ablation of the components of  $S^2$-SAM. It would have been interesting to see the high-sparsity results, as it seems that $S^2$-SAM would have been quite effective there (and if not, why not)? Likewise, an ablation would have been helpful to understand how much we lose by using the previous-gradient version of SAM (the authors do try $S^2$-SAM on dense models, but compare to regular SGD, not regular SAM). Comparing with the original SAM paper, the numbers actually seem pretty promising on dense models, but this is not included in this paper.

I found the sharpness figures confusing. The overall intuition is clear, but the actual process by which the figures were obtained is not.

**Questions:**

How much test accuracy is lost by using the previous-gradient estimate of SAM as compared to the normal SAM?

**Limitations:**

The limitations were adequately addressed.

---

> ### Author Rebuttal · Authors · 2024-08-05
>
> Dear Reviewer UgUE,
>
> Thank you for your review and thoughtful suggestions on our paper. Regarding the questions you raised, we believe they are important points that merit further attention.
>
> **W1: No result on sparsities higher than 90 on ImageNet; how much we lose by using S$^2$SAM compared to regular SAM; no result compared with original SAM on dense model.**
>
> Thank you for your great suggestions. We select our experiment settings of sparsity levels based on existing sparse training papers for easy comparison. To demonstrate S$^2$-SAM is more effective on higher sparsity, we add 95\% sparsity experiments with ResNet-50 on ImageNet. Due to limited time, we adopt RigL and MEST algorithm with S$^2$-SAM. The results are shown in the table below. From the results, we can see that S$^2$-SAM consistently improves the accuracy of sparse training algorithms, and the improvements are similar or higher than 80\% and 90\% sparsity results. Therefore, we can say that S$^2$-SAM is quite effective with high sparsity on ImageNet. We will conduct more experiments with 95\% sparsity on ImageNet and add the results in the revised paper.
>
> | Method | Acurracy at 95\% sparsity |
> | :--- | :---: |
> | RigL | 69.02 |
> | RigL + S$^2$-SAM | **69.83** |
> | MEST(EM) | 69.95 |
> | MEST(EM) + S$^2$-SAM | **70.81** |
>
> To compare with original SAM, we have conducted experiments with sparse training algorithms in Table 4. The reason is that the focus of our paper is on the generalization ability and efficiency of sparse training, instead of dense training. As shown in Table 4, S$^2$-SAM achieves negligible accuracy drop compared to original SAM, while original SAM doubles the computational costs (i.e., 100\% more computations, half training speed). In response to your comment about dense training, we conduct ***additional experiments with SAM on dense training***, and the results are shown in the table below. We can see that dense training with S$^2$-SAM consistently demonstrates accuracy improvements. Compared to original SAM, S$^2$-SAM only experiences negligible accuracy drop, which are completely normal since our method is based on the approximation of sharpness perturbation. We want to stress again that our paper's focus is on a ***practical scenario*** (i.e., sparse training), which means we must consider both generalization ability and the efficiency of the training. Therefore, such small accuracy degradation is totally acceptable since our method has ***zero extra cost*** compared to original SAM which doubles the computation cost. We will integrate those results in our Table 6 and discuss more on this matter in the final version of our paper.
>
> | Method | Original | S$^2$-SAM | SAM |
> | :--- | :---: | :---: | :---: |
> | **CIFAR-10** |  |  |  |
> | ResNet-32 | 94.58 | 94.99 | 95.32 |
> | MobileNet-V2 | 94.13 | 94.55 | 94.77 |
> | VGG-19 | 94.21 | 94.48 | 94.71 |
> |**ImageNet-1K** |  |  |  |
> | EfficientNet-B0 | 76.54 | 77.10 | 77.38 |
> | ResNet-34 | 74.09 | 74.58 | 74.77 |
> | ResNet-50 | 76.90 | 77.32 | 77.58 |
>
> **W2: I found the sharpness figures confusing. The overall intuition is clear, but the actual process by which the figures were obtained is not.**
>
> Thank you for your comments and we are sorry for the confusion. We use the method in citation [11] in our paper to obtain the loss surface visualizations in Figures 1 and 3. The method in citation [11] is a ***widely used*** technique to show the loss surface visualization [R1][R2][R3][R4], employing a random direction method to approximate the 2D projected space of the loss surface. We mention citation [11] in line 44 of our paper to illustrate that higher sparsity indicates a narrower structure, which suggests more chaotic behavior during training, thus degrading accuracy. We will ***further*** explain the method of loss surface visualization in our revised paper, and we will ***cite [11] again in the captions*** of Figures 1 and 3 for clarity.
>
> [R1] Chen, Xiangning, et al. "When Vision Transformers Outperform ResNets without Pre-training or Strong Data Augmentations", ICLR 2022
>
> [R2] Zhang, Xingxuan, et al. "Gradient Norm Aware Minimization Seeks First-Order Flatness and Improves Generalization", CVPR 2023
>
> [R3] Du, Jiawei, et al. "Efficient Sharpness-aware Minimization for Improved Training of Neural Networks", ICLR 2022
>
> [R4] Mi, Peng, et al. "Make Sharpness-Aware Minimization Stronger: A Sparsified Perturbation Approach", NeurIPS 2022
>
> **Q1: How much test accuracy is lost by using the previous-gradient estimate of SAM as compared to the normal SAM?**
>
> Thanks for your question. We compared S$^2$-SAM with SAM in Table 4 for sparse training. We can see from Table 4 that S$^2$-SAM sacrifices only marginal accuracy compared to SAM, but SAM requires about twice the computational cost of S$^2$-SAM or original training. And we also add a more detailed table of comparing the accuracy of SAM and S$^2$-SAM on dense model in the response of W1 (please see the table above) and we will add that to our paper later.

---

> > ### Comment · Reviewer_UgUE · 2024-08-10
> > **Thank you for the rebuttal**
> >
> > Thank you to the authors for the rebuttal and additional experiments. I remain convinced that this paper will be a useful addition to our understanding of SAM and sparse training, and I keep my (positive) score.

---

> > > ### Author Response · Authors · 2024-08-12
> > > **Thank you for your support**
> > >
> > > Dear Reviewer UgUE,
> > >
> > > We sincerely appreciate the time and effort you’ve invested in providing thoughtful and constructive feedback on our submission. We're delighted that you view our work as a valuable contribution to SAM and sparse training. We hope our responses have fully addressed your concerns.
> > >
> > > If you have any further questions, we would be glad to follow up.
> > >
> > > Best regards,
> > >
> > > The Authors

---

### Official Review · Reviewer_Z2wG · 2024-07-11

**Soundness:** 3
**Presentation:** 3
**Contribution:** 3
**Rating:** 7
**Confidence:** 3

**Summary:**

The authors of this paper posit that sparse training is difficult due to a chaotic loss landscape as opposed to standard training of a dense network. In order to address this problem, they propose to perform sparse training with a Sharpness Aware Minimization approach. In order to do so efficiently, they leverage the gradient of the previous iteration to identify the SAM perturbation (S^2-SAM) instead of performing a second forward-backward step like SAM. They show that this approximation of SAM also converges to an optimal solution theoretically.
Extensive empirical evaluations show that the proposed method is effective for sparse training and can also improve robustness of the sparse networks.

**Strengths:**

1. The authors propose a simple method which can be plugged into training any sparse network to improve training via sharpness minimization, without the additional cost of an extra forward pass.
2. Theoretical as well as extensive empirical evaluations are provided to showcase the effectiveness of  S^2-SAM.

**Weaknesses:**

1. The paper starts from the premise that sparse training is difficult in comparison to standard dense training. However, sparse training does not necessarily generalize poorly, in fact some methods are able to train sparse networks that outperform their dense counterparts as shown by Jin et. al. [1] In fact, Jin et. al. claim that pruning can behave as a regularizer, enabling better generalization.
Similarly, Renda et. al. [2] have shown that Learning Rate Rewinding (LRR), a sparsifying algorithm similar to LTs, can find sparse networks that outperform their dense counterparts.
Do these findings suggest that sparse networks can also be found without performing S^2-SAM. Does LRR also find better loss landscapes for sparse networks and if so, then is S^2-SAM necessary? It would also be beneficial to compare with LRR.
2. Does performing SAM instead of the proposed S^2-SAM have a stronger effect on sparse networks i.e. do sparse networks generalize better when trained with SAM instead of S^2-SAM, no matter the training cost. It would be nice to have this comparison to shed light on the robustness of the proposed training method.

**Questions:**

Does the proposed method largely behaves as a regularizer and hence improves generalization of sparse networks? And if so, are there other such regularization methods that could potentially offer similar benefits other than the proposed S^2-SAM. For example, it was suggested by Paul et. al. [3] that as long as the loss landscape is linearly connected for LTs, sparse networks generalize well. Hence, would S2-SAM allow for faster training in this case or potentially pruning a larger fraction of parameters in one iteration in comparison to Iterative Magnitude Pruning?

[1] Jin, Tian, et al. "Pruning’s effect on generalization through the lens of training and regularization." Advances in Neural Information Processing Systems 35 (2022).

[2] Renda, Alex, Jonathan Frankle, and Michael Carbin. "Comparing Rewinding and Fine-tuning in Neural Network Pruning." International Conference on Learning Representations.

[3] Paul, Mansheej, et al. "Unmasking the Lottery Ticket Hypothesis: What's Encoded in a Winning Ticket's Mask?." The Eleventh International Conference on Learning Representations.

**Limitations:**

See above

---

> ### Author Rebuttal · Authors · 2024-08-05
>
> Dear Reviewer Z2wG,
>
> **W1: Some methods are able to train sparse networks that outperform their dense counterparts, do these findings suggest that sparse networks can also be found without performing S$^2$SAM? Is S$^2$SAM necessary? It would be beneficial to compare with LRR.**
>
> Thank you for your constructive comments. We totally agree with you that there are works which can improve the generalization of sparse training. In fact, we will also ***cite those papers and acknowledge their contributions*** to training sparse networks. What we want to stress here and in our paper is that, no matter what sparse training algorithms are used (e.g., LT-based, static sparse training, dynamic sparse training, etc.), the proposed S$^2$-SAM is ***an efficient way to boost*** the performance of such algorithms by improving accuracy with ***no*** extra cost. S$^2$-SAM is not a standalone algorithm, but a ***universal component*** that serves to help any sparse training algorithm achieve better performance. Therefore, no matter how good a sparse training algorithm is, our method will have its ***necessity***. According to Table 1 and Table 2, we already prove that S$^2$-SAM can work well with a variety of sparse training methods. We also conduct LRR and LRR+S$^2$-SAM to further demonstrate its universality. We use ResNet-32 on CIFAR-10 and original training settings, and we report sparsity and accuracy of the IMP process of LRR. From the table below, we can see that LRR+S$^2$-SAM achieves better results compared to LRR, which further proves S$^2$-SAM is a universal applicable method to different kinds of sparse training methods.
>
> | Method | 80\% sparsity | 90\% sparsity | 95\% sparsity |
> | :--- | :---: | :---: | :---: |
> | LRR | 94.68 | 94.05 | 93.82 |
> | LRR + S$^2$-SAM | 94.87 | 94.39 | 94.22 |
>
> **W2: Does performing SAM instead of the proposed S$^2$-SAM have a stronger effect on sparse networks, no matter the training cost?**
>
> Thank you for your question. In fact, we have evaluated different sparse training methods with original SAM and the proposed S$^2$-SAM in Table 4 in our paper. It is true that original SAM achieves slightly better accuracy compared to S$^2$-SAM, but such small accuracy loss is negligible. More importantly, our paper is built upon a very ***practical*** research domain, which is the training efficiency, and our method offers a practical solution for implementing sparse training in ***resource-limited environments***. According to Table 4, traditional SAM doubles the computational costs (i.e., 100\% more computations, half training speed), while our method S$^2$-SAM maintains the benefits of accuracy enhancement while achieving same training speed as original training. Therefore, S$^2$-SAM extends the Pareto boundary by achieving superior outcomes without necessitating compromises.
>
> **Q1: Does the proposed method largely behaves as a regularizer and hence improves generalization of sparse networks? And if so, are there other such regularization methods that could potentially offer similar benefits other than the proposed S$^2$-SAM. Would S$^2$-SAM allow for faster training in this case or potentially pruning a larger fraction of parameters in one iteration in comparison to IMP?**
>
> Thank you for your insightful comments. Regularization generally refers to techniques that improves generalization ability of neural networks by preventing overfitting, such as L1 or L2 regularization that add penalty terms to the loss function to constrain the model's complexity. Therefore, we think SAM-like approaches are also behaving as a regularizer because they are improving the generalization of deep neural networks for various settings. The key idea of SAM-like approaches is to make the model parameters robust to small perturbations in the parameter space, effectively seeking flatter minima. This is achieved by adjusting the parameters to minimize the worst-case loss within a neighborhood around the current parameter values, thus it can be characterized as regularization method. For similar approaches that improve generalization using sharpness of the loss surface, we have conducted a thorough survey and the related literature are cited in our related work section (line 284 - 297). For example, ESAM [13], LookSAM [14] and SAF [15] are all using sharpness information to improve generalization, as well as using specific algorithms to reduce cost. Different from our proposed S$^2$-SAM that has ***zero extra computation cost***, those methods all need extra computation to find a valid perturbation, and they do not target on sparse training.
>
> Applying S$^2$-SAM in Iterative Magnitude Pruning (IMP) with larger pruning fraction of parameter is an intriguing idea. Based on our experiments, S$^2$-SAM enhances the generalization ability of a variety of sparse training methods, including LT (please see Table 1 and Table 2 in our paper). Therefore, S$^2$-SAM may also be able to use fewer LT iterations to achieve good accuracy on similar sparsity. The original LT prunes 20\% remaining parameters at each iteration. We perform additional experiments, which prune 30\%, 40\% parameters at each iteration. The results show when a larger fraction of parameters is pruned, the LT winning ticket accuracy slightly drops. With the proposed S$^2$-SAM, the winning ticket accuracy ***improves*** as expected. We also notice that when the pruning fraction is larger, the effect of S$^2$-SAM is ***more significant***, which proves that S$^2$-SAM successfully solves the difficulty in sparse training. We will conduct more rigorous experiments for future exploration.
>
> | Method | Prune Ratio / Iteration | Iteration | 90\% Sparsity |
> | :--- | :---: | :---: | :---: |
> | LT | 20\% | 11 | 92.31 |
> | LT + S$^2$-SAM | 20\% | 11 | 92.58 |
> | LT | 30\% | 6 | 91.71 |
> | LT + S$^2$-SAM | 30\% | 6 | 92.32 |
> | LT | 40\% | 4 | 90.73 |
> | LT + S$^2$-SAM | 40\% | 4 | 91.44 |

---

> > ### Comment · Reviewer_Z2wG · 2024-08-09
> > **Response to rebuttal**
> >
> > I thank the authors for a detailed explanation and providing additional experiments to highlight the effectiveness of S^2-SAM. I do believe that their proposed method will be useful for sparse training. I have increased my score.

---

> > > ### Author Response · Authors · 2024-08-09
> > > **Thank you for your support**
> > >
> > > Dear Reviewer Z2wG,
> > >
> > > We wanted to express our sincere gratitude for raising our score. Thank you for your support and constructive comments! We will include all the updates in our revision.
> > >
> > > Best regards,
> > >
> > > The Authors

---

### Official Review · Reviewer_uQD3 · 2024-07-12

**Soundness:** 3
**Presentation:** 3
**Contribution:** 3
**Rating:** 6
**Confidence:** 3

**Summary:**

This article introduces S2-SAM (Single-step Sharpness-Aware Minimization), an innovative sharpness-aware optimization method tailored specifically for sparse training with zero extra computational cost.

**Strengths:**

1. The method improves the generalization ability of sparse neural networks, which is a significant challenge in sparse training.

2. Figures of loss surface significantly demonstrate the effectiveness of the proposed methods.

3. S2-SAM provides a general improvement in all the sparse training methods.

**Weaknesses:**

1. S2-SAM seems to be a zero-extra-cost variety of SAM that is implemented in sparse cases. However, when the sparsity is high, the extra cost of SAM can be ignored.

**Questions:**

1. What is the topological initialization of Table 2? Why some of them are non-uniform and some of them are uniform?

2. The datasets in the article are all computer vision domains. Have you done some experiments on NLP domain?

**Limitations:**

This paper doesn't include a limitation section

---

> ### Author Rebuttal · Authors · 2024-08-05
>
> Dear Reviewer uQD3,
>
> We appreciate your review of our paper. The issues you have raised are very important and deserve further discussion. Below are our responses to your comments.
>
> **W1: S2-SAM seems to be a zero-extra-cost variety of SAM that is implemented in sparse cases. However, when the sparsity is high, the extra cost of SAM can be ignored.**
>
> Thank you for your comment. It is true that as sparsity increases, the computation cost decreases for both training and SAM computation. However, we must also consider this scenario in a more ***practical*** perspective, which is the implementation of DNN training on the ***resource-limited devices*** (i.e., the most common case for sparse training). Hardware design usually needs to consider compact footprint to save resources. No matter how much computation saved due to high sparsity, traditional SAM will always double the remaining, leading to at least 100\% increase of the hardware footprint. There remains a significant difference in computational cost between traditional SAM and S$^2$-SAM. Ignoring the additional computation introduced by traditional SAM undermines the purpose of sparse training, which is the ***applicability*** and ***efficiency***.
>
> Our method maintains the benefits of SAM while minimizing computational overhead, offering a practical solution for implementing SAM in resource-limited environments. As shown in Table 4, S$^2$-SAM sacrifices only marginal accuracy compared to SAM, yet SAM requires about ***twice*** the computational cost of S$^2$-SAM or original training. In summary, S$^2$-SAM provides significant advantages in efficiency and applicability across various computational settings, even with high sparsity.
>
> **Q1: What is the topological initialization of Table 2? Why some of them are non-uniform and some of them are uniform?**
>
> The topological initialization refers to the method of distributing non-zero weights across the network layers, which is presented in two ways in Table 2: uniform and non-uniform.
>
> Uniform: The sparsity $s^l$ of each individual layer is equal to the total sparsity $S$ of the network.
>
> Non-uniform (ERK): The number of parameters in the sparse convolutional layers is scaled proportionally to the width and height of the $l^{th}$ convolutional kernel.
>
> These two sparsity distributions are widely used in current research [R1][R2][R3]. The reason we use these two sparsity distributions in Table 2 is to provide a fair and comprehensive comparison with other methods.
>
> [R1] Evci U, Gale T, Menick J, et al. "Rigging the lottery: Making all tickets winners", ICML 2020
>
> [R2] Liu, Shiwei, et al. "Do we actually need dense over-parameterization?", ICML 2021
>
> [R3] Yuan, Geng, et al. "Mest: Accurate and fast memory-economic sparse training framework on the edge", NeurIPS 2021
>
> **Q2: The datasets in the article are all computer vision domains. Have you done some experiments on NLP domain?**
>
> Yes, we evaluate our methods on a translation task using the Transformer model [R4] on WMT-14 En-De dataset, reporting the best SacreBLEU scores on the validation dataset in the table below. We applied our method with uniform sparsity levels of 80\% and 90\% across all layers. Compared to MEST(EM), our method demonstrates improved performance in both dense and high-sparsity scenarios. We will also integrate more NLP results in our revised paper.
>
> | Method | SacreBLEU |
> | :--- | :---: |
> | Dense | 27.6 |
> | Dense + S$^2$-SAM | 27.9 |
>
>
> | Method | SacreBLEU | SacreBLEU |
> | :--- | :---: | :---: |
> |  | 80\% sparsity | 90\% sparsity |
> | MEST(EM) | 27.1 | 26.4 |
> | MEST(EM) + S$^2$-SAM | 27.5 | 27.2 |
>
> [R4] Vaswani A, et al. "Attention is all you need", NeurIPS 2017

---

### Official Review · Reviewer_ZjXc · 2024-07-14

**Soundness:** 3
**Presentation:** 3
**Contribution:** 4
**Rating:** 7
**Confidence:** 3

**Summary:**

This paper studies the challenges of training sparse neural networks directly and identifies one of the contributing factors, i.e., the chaotic loss surface. Consequently, it proposes a new method, i.e., Single-step Sharpness-Aware Minimization (S2-SAM), tailored specially to train sparse networks. S2-SAM is based on SAM for dense neural networks training with the main difference that it uses just one gradient computation (thus, more efficient), while SAM uses two gradient computations. Experimental results show unanimously that S2-SAM improves the performance of all sparse training methods studied.

**Strengths:**

* Original paper idea with well-designed execution
* The paper is easy to read and follow
* Novel proposed method with theoretical flavour
* Well designed empirical validation
* Impressive boost in performance for all sparse training methods studied when the proposed method is applied to them
* The paper is significant for the sparse training community and has the potential of changing how sparse training methods are designed nowadays

**Weaknesses:**

* I don’t see major weak points, except that it is not clear when the source code will be made available for easy reproducibility

**Questions:**

Q1) It seems that S2-SAM works with sparse networks, but not too well with dense networks. As sparser the network is as more impactful S2-SAM is on the overall performance (lines 252-253 and the majority of the results). Can you prepare a systematic study (experiment) to quantify and illustrate better this behavior (e.g., by varying the sparsity level from 0 to 100% in small steps, or alternatives…)?

Q2) Could you present in an Appendix how the loss surface visualisations from Figures 1 and 3 have been computed? I see that you cite [6], but overall this seems to be an approximation method to visualize the loss surface of a very high dimensional space. If so, this shall be properly acknowledged to avoid bringing in readers’ mind inaccurate ideas.

Q3) It seems that on ImageNet, which is a much more challenging dataset, your proposed method together with MEST or RigL outperforms the dense baseline (without S2-SAM). Can you present the results on this latter case also? Do you have any idea why this behaviour is different on CIFAR 10/100? Probably, this deserves a longer qualitative discussion.

Q4) The whole empirical validation has been performed on convolutional neural networks (not on all neural network architectures), and the “chaotic” loss surface is probably just one (not the only one) of the reasons which can help understand the training behaviors of sparse neural networks. Can you please identify better in the paper the boundaries of your work to avoid inaccurate conclusions in the readers’ mind?

Q5) (minor) To expand the broadness of the experimental results. In Table 1, would it be possible to add also the 80% sparsity level?In Table 2, would it be possible to add also a SET 5x?

---

> ### Author Rebuttal · Authors · 2024-08-05
>
> Dear Reviewer ZjXc,
>
> We sincerely appreciate your thoughtful comments. All source code will be released after the paper is accepted.
>
> **Q1: Systematic study to quantify and illustrate the proposed method?**
>
> Thank you for the question. We want to stress that our paper is focusing on providing a universal solution for training a sparse neural network with different algorithms. Compared to dense training, our contribution has more ***practical*** significance. Meanwhile, our method works fine on dense training (negligible accuracy loss against SAM) with a significant speed improvement compared to SAM. The table below shows the comparison between MEST with and without our proposed method, S$^2$-SAM, in terms of accuracy on ResNet-32 under different sparsity levels from 0% to 98%.
>
> | Sparsity | Dense | 20\% | 40\% | 60\% | 80\% | 90\% | 95\% | 98\% |
> | :---: | :---: | :---: | :---: | :---: | :---: | :---: | :---: | :---: |
> | MEST(EM) | 94.58 | 94.35 | 93.92 | 93.58 | 93.23 | 92.56 | 91.15 | 89.22 |
> |MEST(EM) + S$^2$-SAM| 94.99 | 94.67 | 94.38 | 94.11 | 93.88 | 93.43 | 91.58 | 91.22 |
> | $\Delta$ Accuracy | 0.41 | 0.32 | 0.46 | 0.53 | 0.65 | 0.87 | 0.43 | 2.00 |
>
> As shown in the table, as sparsity increases, the accuracy difference between MEST with and without S$^2$-SAM becomes more significant. We will include this table in the revised version of our paper.
>
> **Q2: Loss surface visualizations method and citation [6].**
>
> Thank you for your question and sorry for the confusion we made. In our paper, we used the method cited as [11] to generate the loss surface visualizations in Figures 1 and 3, and we use the metric in [6] to compute the sharpness of the surface. We will make it clear in our revised paper.
>
> The method in citation [11] is a ***widely used*** technique for showing loss surface visualizations [R1-R4], employing a random direction approach to approximate the 2D projected space of the loss surface (i.e., 2D contour with loss values which can be converted to 3D image). We will further explain the method of loss surface visualization in our revised paper, and we will cite [11] again in the captions of Figures 1 and 3 for clarity.
>
> From citation [6] in our paper, we identify the $Ra$ value in the captions of Figures 1 and 3 as the mean absolute deviation of the z-axis value (loss value) to evaluate the sharpness of the surface. A smaller $Ra$ indicates a smoother loss surface, suggesting improved generalization ability.
>
> We will explain with more details of [6] and [11] in the Appendix in the revised version of our paper.
>
> [R1] Chen, Xiangning, et al."When Vision Transformers Outperform ResNets without Pre-training or Strong Data Augmentations", ICLR 2022
>
> [R2] Zhang, Xingxuan, et al. "Gradient Norm Aware Minimization Seeks First-Order Flatness and Improves Generalization", CVPR 2023
>
> [R3] Du, Jiawei, et al. "Efficient Sharpness-aware Minimization for Improved Training of Neural Networks", ICLR 2022
>
> [R4] Mi, Peng, et al. "Make Sharpness-Aware Minimization Stronger: A Sparsified Perturbation Approach", NeurIPS 2022
>
> **Q3: Why some sparse training accuracy higher than dense baseline on ImageNet but not on CIFAR?**
>
> The reason for this behavior is that the CIFAR datasets are relatively small, which can make DNNs prone to overfitting and makes it harder to improve accuracy significantly. In contrast, the ImageNet dataset is significantly larger, providing a more challenging and diverse set of images that help in training more generalized models that conquer overfitting. Under such circumstances, extending training time (***MEST 1.7$\times$*** or ***RigL 5$\times$*** in Table 2) will be very helpful to improve accuracy as the original accuracy (MEST or RigL without S$^2$-SAM) is already very close to the dense training baseline. We will provide this discussion in the revised paper.
>
> **Q4: Identify in the paper the boundaries of your work regarding tasks.**
>
> Thank you for the good suggestion. Our paper mainly explores the CNN structures and their loss surfaces to understand the training behaviors of sparse neural networks. We will emphasize this focus in the Introduction part in the revised version of the paper to ensure clarity and avoid any inaccurate conclusions.
>
> We also conduct additional experiments on the NLP task due to reviewer's comment. We evaluate our methods on a translation task using the Transformer model [R5] on WMT-14 En-De dataset, reporting the best SacreBLEU scores on the validation dataset in the table below. We applied our method with uniform sparsity levels of 80\% and 90\% across all layers. Compared to MEST(EM), our method demonstrates improved performance in both dense and high-sparsity scenarios. We will integrate more NLP results in our revised paper.
>
> | Method | SacreBLEU |
> | :--- | :---: |
> | Dense | 27.6 |
> | Dense + S$^2$-SAM | 27.9 |
>
> | Method | SacreBLEU | SacreBLEU |
> | :--- | :---: | :---: |
> |  | 80\% sparsity | 90\% sparsity |
> | MEST(EM) | 27.1 | 26.4 |
> | MEST(EM) + S$^2$-SAM | 27.5 | 27.2 |
>
> [R5] Vaswani A, et al. "Attention is all you need", NeurIPS 2017
>
> **Q5: Add 80% sparsity in Table 1 and add SET 5x in Table 2.**
>
> Thank you for your thoughtful suggestions. We report 90\%, 95\%, and 98\% sparsity in our paper at first place because most current literature primarily utilizes them for comparison. Due to time and resource limits, we report the accuracy at 80\% sparsity on CIFAR-10 in the response of Q1 for Table 1, and we will include that and all the other accuracy result in 80\% sparsity in the revised version of our paper to guarantee a more comprehensive comparison.
>
> For Table 2, we plan to include the results from SET 5$\times$ in a future revision. And below is the table for the comparison of SET 5$\times$ with and without our method on ImageNet dataset at 80% and 90% sparsity respectively.
>
> | Method | 80\% sparsity | 90\% sparsity |
> | :--- | :---: | :---: |
> | SET $_{5 \times}$ | 74.60 | 72.43 |
> | SET $_{5 \times}$ + S$^2$-SAM | 75.43 | 73.16 |

---

> > ### Comment · Reviewer_ZjXc · 2024-08-12
> > **Rebuttal acknowledgement**
> >
> > Dear authors,
> >
> > Thank you for considering my comments and for the well prepared rebuttal. I will keep my original score (accept).
> >
> > Best wishes,

---

### Decision · Program_Chairs · 2024-09-25

**Decision:**

Accept (poster)

**Comment:**

This paper proposes the application of Sharpness-Aware Minimization (SAM) to sparse training to enhance its generalization performance. To maintain the low cost of sparse training, the paper introduces a variant of SAM that requires only a single gradient computation by approximating sharpness perturbation using prior gradient information. Extensive empirical results are presented across various sparse training approaches and evaluation metrics. All reviewers agree that this paper makes a significant contribution to the field of sparse training.